# Lipid-protein interactions modulate the conformational equilibrium of a potassium channel

Ruo-Xu Gu [1] & Bert L. de Groot[1✉]

Cell membranes actively participate in the regulation of protein structure and function. In this work, we conduct molecular dynamics simulations to investigate how different membrane environments affect protein structure and function in the case of MthK, a potassium channel. We observe different ion permeation rates of MthK in membranes with different properties, and ascribe them to a shift of the conformational equilibrium between two states of the channel that differ according to whether a transmembrane helix has a kink. Further investigations indicate that two key residues in the kink region mediate a crosstalk between two gates at the selectivity filter and the central cavity, respectively. Opening of one gate eventually leads to closure of the other. Our simulations provide an atomistic model of how lipid-protein interactions affect the conformational equilibrium of a membrane protein. The gating mechanism revealed for MthK may also apply to other potassium channels.

[1] Department of Theoretical and Computational Biophysics, Max-Planck Institute for Biophysical Chemistry, Am Fassberg 11, 37077 Göttingen, Germany.
✉email: bgroot@gwdg.de

Cell membranes not only provide a matrix for various biochemical reactions mediated by membrane proteins, but also actively participate in regulating the structure and function of membrane proteins. How lipid–protein interactions regulate protein structure and function[1–3] is of great interest and has been widely discussed. The topic is even more intriguing considering the spatially and temporally heterogeneous mixing of the cell membranes[4–6]. Both specific lipid–protein interactions and general membrane properties are known to play a crucial role in regulating protein structure and function[3]. For instance, general membrane properties, including bilayer thickness and lateral pressure, are able to regulate the equilibria of protein conformations. Corradi et al.'s work[7] provides a comprehensive review of experiments and molecular dynamics (MD) simulations of how lipid–protein interactions affect protein structure and function.

In this work, we investigate how lipid–protein interactions regulate the conformation and function (ion permeation) of potassium channels. Potassium channels are of great importance due to their prevalence in all living organisms and their fundamental role in multiple biological processes[8]. Extensive experimental studies have suggested the sensitivity of the structure and function of these channels to the membrane environment (see Supplementary Note 1). For example, larger lateral stress in thicker membranes is observed to reduce the open probability and increase the energy barrier of conformational changes between the open and closed states of potassium channels, such as the large-conductance $Ca^{2+}$-activated $K^+$ channel ($BK_{Ca}$)[9,10] and KcsA[11].

However, it remains unclear how the channel structure responds to the membrane environment at the atomistic level. This is where MD simulations can provide valuable insights. In this work, we focused on the interactions between lipids and the pore domain. The pore domain constitutes the ion permeation pathway, and its sequence and quaternary structure are conserved among different potassium channels. We employed a potassium channel from *Methanothermobacter thermautotrophicus*, MthK, as a reference, due to the extensive available experimental studies of this channel and the relatively simple structure of its transmembrane domain. The pore domain of MthK, depicted in Fig. 1, comprises four subunits, each containing two transmembrane helices that are connected by a pore helix and a loop with the conserved signature amino acid sequence: Thr–Val–Gly–Tyr–Gly. The conserved loop regions form a selectivity filter with four potassium-binding sites at the extracellular half of the membrane. The selectivity filter is also known as C-type gate or inactivation gate for some potassium channels[12–14]. Four inner helices surround a central cavity beneath the selectivity filter, which works as the activation gate[15,16]. The selectivity filter and the central cavity constitute the ion permeation pathway, and the interplay between them modulates ion conduction.

To test the effects of several general membrane properties, we simulated the pore domain of MthK[17], embedded in 11 different membrane environments. We investigated the function (ion currents) of MthK in the presence of a transmembrane voltage. We indeed identified different ion permeation rates in different lipid environments (i.e., membrane thickness, saturation degree of lipid tails, and cholesterol content), consistent with experimental studies[9,10,18,19]. The differential permeation rates are the consequence of the equilibrium shift between two conformations of the inner helix. More importantly, the conformational changes of the inner helix revealed by our simulations define a crosstalk between two gates at the selectivity filter and the central cavity, respectively.

## Results

**Membrane properties regulate the ion permeation rate of MthK.** The lipids used in this work are divided into four categories: (a) lipids with two saturated tails (DLPC, DMPC, and DPPC), (b) lipids with two unsaturated tails (DYPC, DOPC, DGPC, DEPC, and DNPC), (c) hybrid lipids (one saturated tail and one unsaturated tail, POPC), and (d) mixtures of PC lipids and cholesterol (DLPC:cholesterol and POPC:cholesterol mixtures). Ion currents are plotted as a function of membrane thickness (Fig. 2). A complete list of the data is shown in Supplementary Table 2.

Ion currents change as a function of bilayer thickness (see the black circles and gray squares in Fig. 2). For membranes composed of saturated lipids (lipid category (a), see the black circles in Fig. 2), the ion permeation rate in DMPC (6.2 pA) is higher than those in both thinner (DLPC) and thicker (DPPC) membranes (4.2 and 3.8 pA, respectively). For membranes composed of unsaturated lipids, a larger current is obtained in DYPC and DOPC (3.5 and 4.6 pA, respectively), whereas thicker membranes (DGPC, DEPC, and DNPC) reduce ion conduction dramatically (~1 pA, lipid category (b), see the gray squares in Fig. 2). An ion current of 6.3 pA is observed in POPC (lipid category (c), red triangle in Fig. 2), whose thickness is comparable with DMPC and DOPC. We conclude that ion permeation

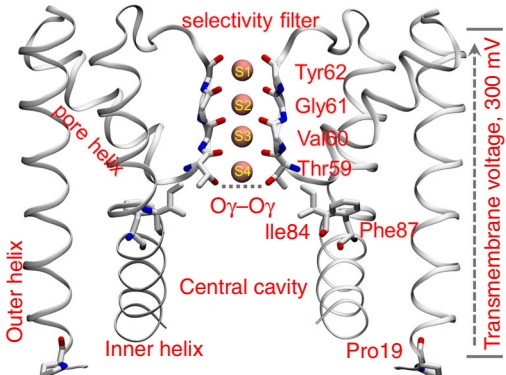

**Fig. 1 Structure of the MthK pore domain.** Only two subunits of the tetramer are shown for clarity. The four potassium-binding sites (pink dots) in the selectivity filter are labeled as S1–S4. The distance between Oγ atoms, the oxygen atoms of the Thr59 side chains, is depicted as the dashed line. Other residues that play a critical role in gating and ion permeation are labeled accordingly.

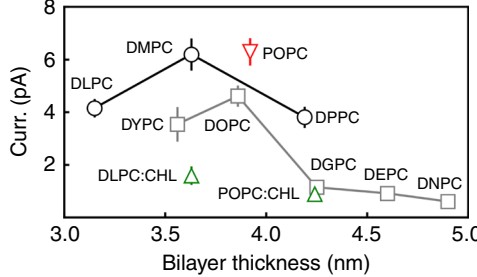

**Fig. 2 Effects of lipid–protein interactions on ion permeation through MthK.** Ion currents are shown as a function of membrane thickness. Ion currents of MthK in membranes composed of saturated (DLPC, $n = 40$; DMPC, $n = 30$; DPPC, $n = 40$), unsaturated (DYPC, $n = 20$; DOPC, $n = 40$; DGPC, $n = 25$; DEPC, $n = 20$; DNPC, $n = 10$), and hybrid lipids (POPC, $n = 40$), and cholesterol-containing membranes (DLPC:CHOL, $n = 30$; POPC:CHOL, $n = 10$) are shown in black, gray, red, and green symbols, respectively. $n$ indicates the number of simulation replicates. Data are presented as mean values ± SEM. Source data are available as a Source Data file.

reaches an optimum with a bilayer thickness ranging from ~3.5 to ~4.0 nm, whereas both smaller and larger thickness reduce the ion conduction rate.

A larger unsaturation degree of lipid tails reduces the ion permeation rate, as suggested by smaller currents in membranes composed of unsaturated lipids, than in bilayers composed of saturated or hybrid lipids (compare lipid categories (a) and (b), the black and gray lines in Fig. 2). We compared currents in bilayers with similar thickness to rule out an indirect effect of membrane thickness: ion conduction in DOPC was slower than that in DMPC and POPC (4.6 pA vs. 6.2 and 6.3 pA), despite their similar thickness (3.9, 3.6, and 3.9 nm, respectively); the same trend is observed by comparing DGPC with DPPC (thickness of 4.2 nm, currents of 1.2 and 3.8 pA), as shown in Fig. 2.

Cholesterol-containing membranes reduce ion conduction (Fig. 2). We tested two cholesterol-containing membranes (lipid category (d), see the green triangles in Fig. 2) and compared the resulting currents to cholesterol-free membranes (see the black circles and red triangle in Fig. 2): ion current in a DLPC: cholesterol mixture is smaller than that in the DMPC bilayer (1.6 vs. 6.2 pA); similarly, ion current in a POPC:cholesterol mixture is smaller than that in DPPC (0.9 vs. 3.8 pA). Again, these comparisons ruled out the effects of membrane thickness. We observed depletion of cholesterol from the protein in our simulations, despite cholesterol–MthK interactions occasionally found in some of the trajectories. We propose that cholesterol may modulate ion conduction of MthK by changing the lateral pressure of the membrane, instead of specific lipid–protein interactions.

In summary, a variety of membrane properties, including membrane thickness, unsaturation degree of lipid tails, and cholesterol content, affect ion conduction through MthK.

**Conformational equilibrium of MthK regulates ion conduction**. The two transmembrane helices are the portion of the protein in direct contact with lipids. Hence, we investigate their conformational changes in different membranes to understand how lipid–protein interactions affect the actual ion permeation rate.

Crystal structures of MthK show a kink of the inner helix in the vicinity of Gly83, just below the selectivity filter in the quaternary structure (Fig. 3a). This kink is characterized by a break of the Val81–Gly85 and Leu82–Thr86 backbone hydrogen bonds of the α-helix (indicated by dashed lines in Fig. 3a). A water molecule binds to the backbone atoms of this region in crystal structures, which may stabilize the kink (Fig. 3a). This state is used as the initial conformation in our simulations. However, in our simulations, the helical backbone is able to regain the hydrogen bonds when the water molecule occasionally diffuses away and adapts to another state without this kink. These two states also differ in their bending angle (~43° and ~30° for the state with and without the kink, respectively, Fig. 3a). We describe the kink by the distance between the backbone oxygen atom of Val81 and the amide hydrogen atom of Gly85 (referred to as Val81O–Gly85HN distance). The histogram (Fig. 3b) as a function of the Val81O–Gly85HN distance and bending angle clearly indicates the existence of the above-mentioned two states in simulations. These two states are referred to as kinked and bent states in the following.

The kinked state is more prevalent in simulations, but the bent state is associated with permeation events. As shown in Fig. 3c, ion currents positively correlate with the fraction of the bent state. We therefore hypothesize that the kinked state may inhibit ion conduction, and a transition to the bent state facilitates ion permeation. The systems we employed can be categorized into three groups based on ion currents: (1) DMPC and POPC, with

ion currents of ~6 pA; (2) DLPC, DYPC, DOPC, and DPPC, with ion currents ranging from 3.5 to 4.6 pA; (3) DGPC, DEPC, DNPC and cholesterol-containing membranes, with conduction rates <2 pA. The fraction of bent states in these three categories is ~30%, ~10–15%, and <4%, respectively. For each system in the second group, we divide the simulation replicates into two categories based on the prevalence of the bent state in each replicate. The categories with higher average bent-state fractions (~25%) present larger ion currents (~6 pA) than the categories with lower bent-state fractions (<7%, currents range from 1.5 to 3.5 pA) (Fig. 3d, Supplementary Table 3). In addition, restraining conformations of all four subunits to the kinked state in POPC inhibits ion conduction (ion current is reduced from 6.3 pA in unrestrained simulations to 0.4 pA, Fig. 3e, Supplementary Table 4). In contrast, restraining one of the four subunits in the bent state in DOPC (bent-state fraction increases from ~10% in unrestrained simulations to 27%) increases the current from 4.6 to 5.9 pA. We also restrained all of the four subunits in the bent state in the DOPC membrane. However, a decrease in ion current is observed in this case (2.9 pA), due to dehydration of the central cavity (see below).

Note that the protein is a tetramer, so the bent-state fraction is 25% if one subunit of the tetramer protein is in the bent state. In our simulations, ion permeation is observed for an asymmetrical conformation of MthK, in which one subunit is in the bent state, whereas the other three are in the kinked state. The channel is nonconductive if all of the four subunits are in the kinked state.

We determine the motion of the outer helix by measuring the distance between the Cα atoms of Pro19 (a residue at the end of the outer helix, Fig. 1) of different subunits. We observe smaller Pro19 Cα–Cα distances in thicker membranes (Fig. 3f and Supplementary Table 2), suggesting conformational changes of the outer helix. We also observed a correlation between the Pro19 Cα–Cα distance and ion permeation rate (Supplementary Tables 2 and 4, see Supplementary Note 2 for detailed description). We conclude that conformational changes of the outer helices are also involved in modulating ion permeation. We further propose that a smaller Pro19 Cα–Cα distance may reduce the ion permeation rate by stabilizing the kinked state of the inner helix, as indicated by the coupling between the conformations of the two helices (Fig. 3g, Supplementary Figs. 1 and 2, Supplementary Tables 2–4, and Fig. 4a). See Supplementary Note 3 for a detailed discussion of the conformational coupling between the two helices.

**Gating of the selectivity filter by the inner helix**. As the permeation pathway, including the selectivity filter and the central cavity, is the portion of the protein that is in direct contact with potassium during the ion conduction process, we studied how the kinked and bent states regulate its structure, to interpret the correlation between protein conformational equilibrium and ion conduction rates.

The conformation of the inner helix modulates the opening degree of the selectivity filter, which is the main barrier for ion permeation in simulations using different membranes. Two extreme conformations corresponding to kinked (blue) and bent (red) states revealed by FMA are shown in Fig. 4a. FMA implies a closing motion of the selectivity filter when the inner helix changes from the bent to the kinked state (the motion is indicated by arrows in Fig. 4a).

To give a quantitative description, we determine the opening degree of the selectivity filter by measuring the distance between the side-chain Oγ atoms of Thr59, the first residue constituting the selectivity filter (see Fig. 1), as in Kopec et al.'s work[20]. In simulations using different membranes, the Thr59 Oγ–Oγ distance positively correlates with the bent-state fraction of the

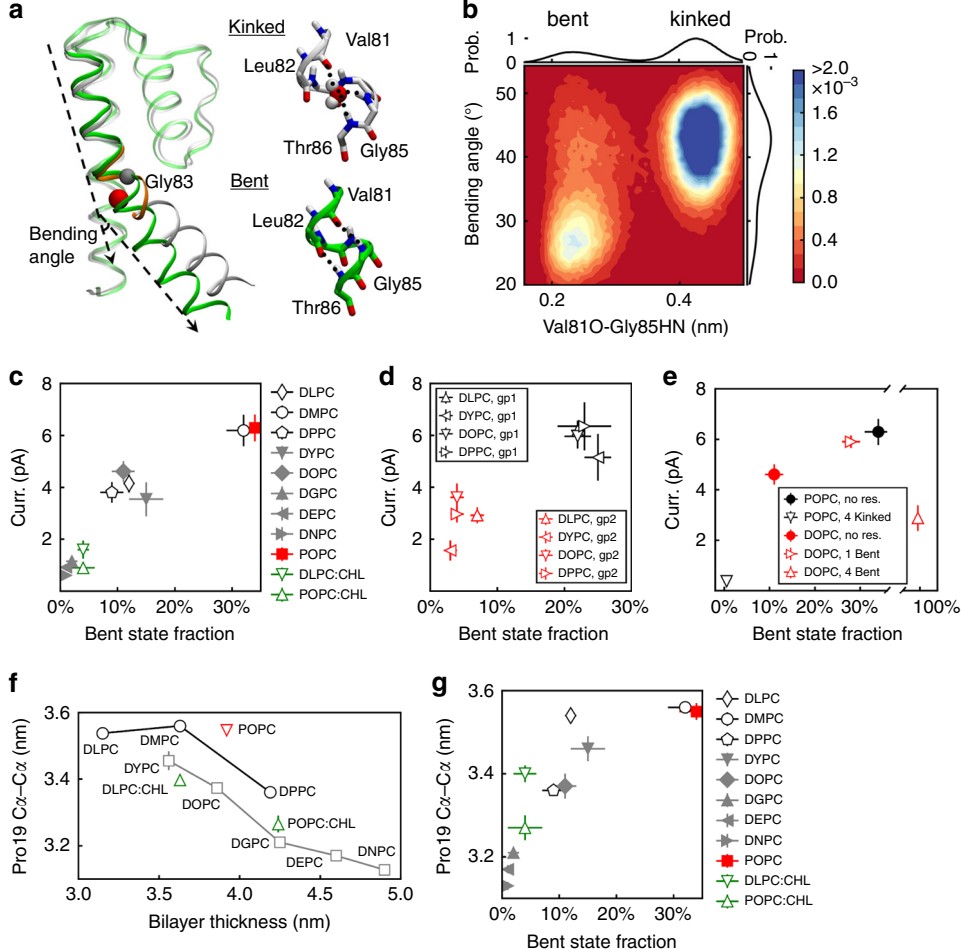

**Fig. 3 Equilibrium between the bent and the kinked states modulates the ion permeation rate. a** Superimposition of the kinked (white) and the bent (green) state. The kink region is highlighted in orange. Bending angle of the inner helix is indicated by dashed arrows. The Cα atom of Gly83 and the water molecule bound to the backbone in the kinked state are shown in gray and red spheres, respectively. A close-up view of the kink region is shown for the two states. Potential hydrogen bonds are indicated by dotted lines. **b** Histogram as a function of the bending angle of the inner helix and the Val81O–Gly85NH distance. The results of simulations in the POPC membrane are shown. Average ion current as a function of bent-state fraction of the inner helix for **c** simulations using different membranes; **d** simulations in DLPC, DYPC, DOPC, and DPPC membranes; simulation replicates are divided into two groups (labeled as gp1 and gp2, respectively), and the average currents and bent-state fractions are calculated separately for the two groups; **e** simulations with inner helices restrained in the kinked and the bent states, respectively (open symbols). Simulations in the corresponding membranes without restraints are also shown for comparison (filled symbols). **f** Pro19 Cα–Cα distance as a function of membrane thickness. **g** Coupling of the conformations of the inner (characterized by the bent-state fraction) and outer (characterized by the Pro19 Cα–Cα distance) helices. Number of simulation replicates for each data point in panels (**c**), (**f**), and (**g**) is the same with Fig. 2. The results of DLPC, DYPC, DOPC, and DPPC bilayers in panel (**d**) include 25, 9, 23, and 30 independent replicates for gp1, and 15, 11, 17, and 10 replicates for gp2. Each data point of the restrained simulations in panel (**e**) is an average of 10 independent replicates. Data are presented as mean values ± SEM. Source data are available as a Source Data file.

inner helix (Fig. 4b). In simulations restraining four subunits in the kinked state, one subunit, or all four subunits in the bent state, the Thr59 Oγ–Oγ distances are 0.497, 0.512, and 0.533 nm, respectively (Supplementary Table 4). These data verify the conclusion from FMA that the bent state induces a larger opening degree of the selectivity filter. The relevance of the opening degree of the selectivity filter to ion permeation rate is confirmed by the correlation between the Thr59 Oγ–Oγ distance and the ion current in simulations using different membranes with and without restraints (Supplementary Fig. 3), in agreement with Kopec et al.[20]. Note that the simulation restraining the four subunits in the bent state is an exception, as a small current (2.9 pA) is observed for a very large opening of the selectivity filter (0.533 nm, Supplementary Fig. 3). This exception is due to dehydration of the central cavity (see below).

Further investigation suggests that the selectivity filter opening is not only facilitated by the local expansion of the intracellular entrance (determined by Thr59 Oγ–Oγ distance), but also a global motion of the whole selectivity filter (the "free energy" profiles in Fig. 4c, correlation between Thr59 Oγ–Oγ distance and the Thr59 and Gly61 backbone oxygen distances in Fig. 4d, potassium occupancies in Supplementary Fig. 5). See Supplementary Note 4 for a detailed discussion.

Interactions between the side chains of Ile84, a residue in the kink region, and Thr59, the residue constituting the S4 potassium-binding site, mediate conformational changes of the inner helix to different opening degrees of the selectivity filter. Specifically, the bent state moves the Ile84 away from Thr59, as suggested by FMA (Fig. 4a). The Thr59 residues of the four subunits are then pulled away from each other by Ile84–Thr59

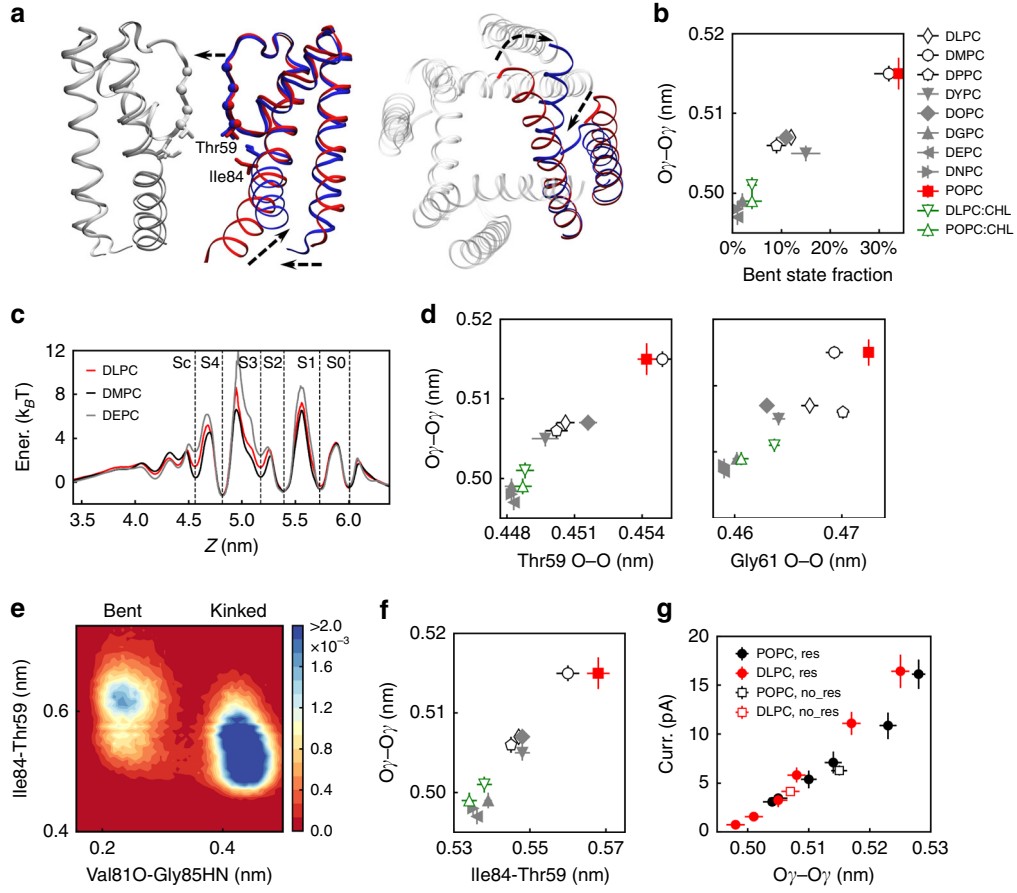

**Fig. 4 Gating of the selectivity filter by the kink of the inner helix.** Panels (**b**), (**d**), and (**f**) share the same legend. **a** Side (left) and top (right) view of two extreme conformations from functional mode analysis (FMA), which correspond to the kinked (blue) and the bent (red) states. Directions of conformational changes are indicated by dashed arrows. The subunit whose bending angle was used to decode the collective motions is shown in red and blue, whereas the other subunits are depicted in gray. **b** Correlation between Thr59 Oγ–Oγ distance and bent-state fraction of the channel in simulations using different membranes. **c** Potassium "free energy" profiles along the ion permeation pathway in DLPC (red), DMPC (black), and DEPC (gray) membranes. Potassium-binding sites are indicated with dashed lines. **d** Correlation between Thr59 Oγ–Oγ distance and the backbone oxygen atom distance of Thr59 (left panel) and Gly61 (right panel) in simulations using different membranes. **e** Histogram as a function of the Ile84–Thr59 side-chain distance and the Val81–Gly85HN distance. **f** Thr59 Oγ–Oγ distance as a function of the Ile84–Thr59 side-chain distance in simulations using different membranes. **g** Ion current as a function of the Thr59 Oγ–Oγ distance in simulations restraining the Ile84 side-chain distances in POPC and DLPC membranes (filled symbols). Data from unrestrained simulations are also shown for comparison (open symbols). Number of independent replicates for each data point in panels (**b**), (**d**), and (**f**) is the same with Fig. 2. Ten independent simulations were performed for each data point of the restrained simulations in panel (**g**), except for the simulation in POPC with an average Thr59 Oγ–Oγ distance of 0.528 nm, which includes 20 independent replicates. Data are presented as mean values ± SEM. Source data are available as a Source Data file.

side-chain interactions and the Thr59 Oγ–Oγ distance is increased.

The analysis of simulation trajectories confirms different Ile84–Thr59 distances in the bent and kinked states. The 2D histogram as a function of the Val81O–Gly85HN distance and the Ile84–Thr59 side-chain distance reveals a smaller value of the Ile84–Thr59 side-chain distance in the kinked sate (Fig. 4e, average values of 0.54 and 0.59 nm for the two states, respectively). In addition, simulations restraining the four subunits in the kinked state, one subunit, or all four subunits in the bent state reveal Ile84–Thr59 side-chain distances of 0.53, 0.56, and 0.64 nm, respectively (Supplementary Table 4). The importance of Ile84–Thr59 side-chain interactions for the opening degree of the selectivity filter is indicated by the following analyses and simulations: (a) a strong correlation between the Thr59 Oγ–Oγ distance and the Ile84–Thr59 side-chain distance is observed (Fig. 4f, Supplementary Fig. 4); (b) mutation of Ile84 to Ala, a residue with a smaller side chain, reduces the effects of inner helical conformation on the selectivity filter opening: in

simulations of the mutant in POPC and DLPC membranes, a smaller bent-state fraction in DLPC membranes (16% vs. 33%, similar to the wild type, Supplementary Table 2) does not lead to smaller Oγ–Oγ distances (0.513 vs. 0.515 nm, respectively), as the wild type did (corresponding values are 0.507 and 0.515 nm, Supplementary Table 2); (c) controlling the distance between the centers of mass (COM) of Ile84 side chains of the four subunits modifies the opening degree of the selectivity filter (i.e., Thr59 Oγ–Oγ, see Fig. 4g and Supplementary Table 5), and results in similar ion currents at given Oγ–Oγ distances in POPC and DLPC membranes (Fig. 4g, Supplementary Table 5). These simulations strongly indicate that gating of the selectivity filter is the direct reason for different ion permeation rates in different membrane environments.

Ile84–Thr59 interactions locally occur at the contacts between the selectivity filter and the inner helices, and mainly affect the opening degree of the intracellular entrance of the selectivity filter (Thr59 Oγ–Oγ distance). The aforementioned coupling motion of the selectivity filter residues (Fig. 4c, d) is ascribed to a global

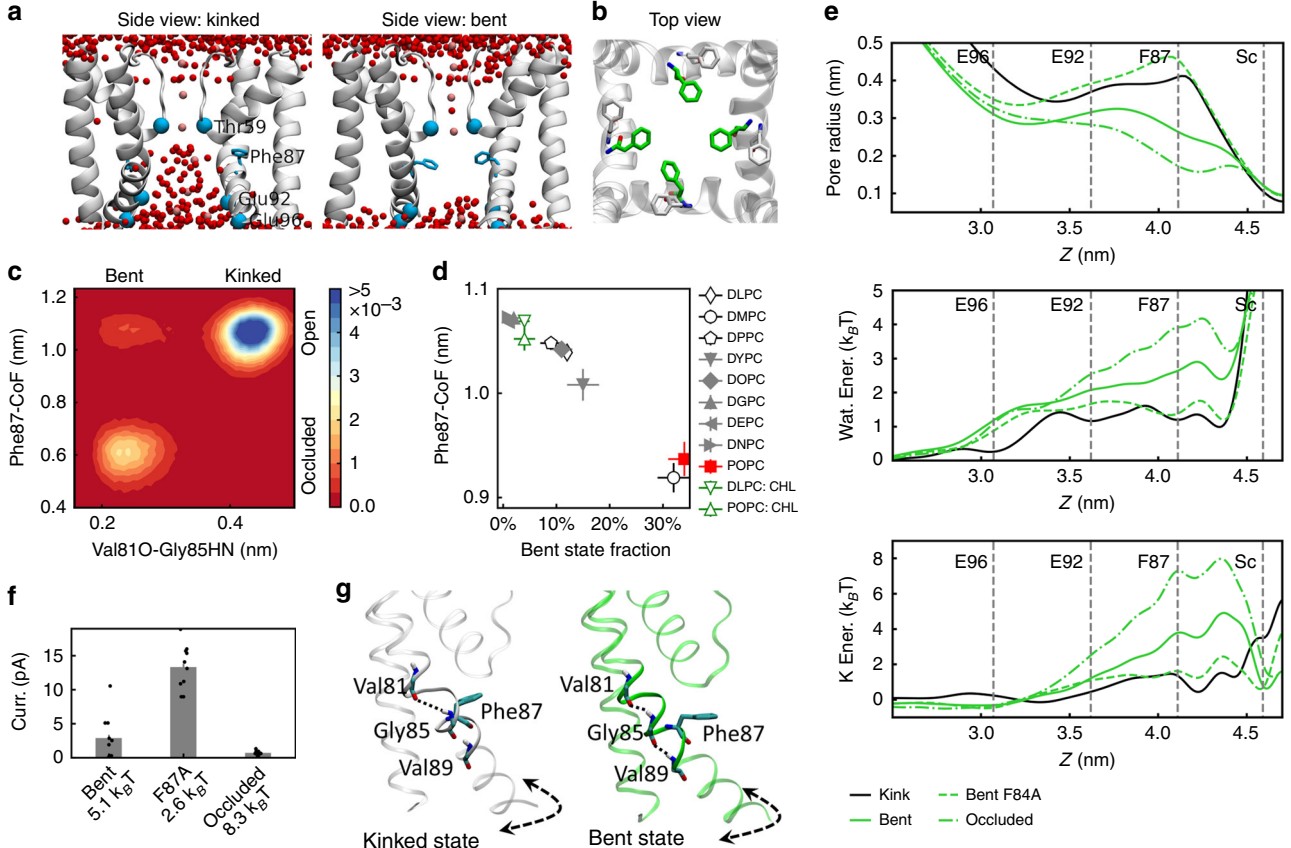

**Fig. 5 Gating of the central cavity by the kink of the inner helix. a** Different Phe87 side-chain orientations in the kinked (left half) and bent (right half) states result in different hydration degrees of the central cavity. Water molecules and potassium ions are shown as red and pink spheres, respectively. Phe87 is shown as cyan stick model. Cα atoms of Thr59, Glu92, and Glu96 are shown as cyan spheres. **b** Top view of Phe87 side-chain orientation in the kinked (white) and bent (green) state. The inner helices are also shown in white. **c** Histogram as a function of the Val81O–Gly85HN distance and the Phe87-CoF distance (result of simulations in POPC membranes). **d** Phe87-CoF distance as a function of the bent-state fraction. **e** Pore radius (top), "free energy" profiles of water (middle), and potassium (bottom) in simulations with different dehydration degrees of the central cavity. Results of the kinked state, the bent state, the bent state including the Phe87Ala mutation, and the occluded state are shown. Positions of Glu96, Glu92, and Phe87, as well as the Sc site, are indicated by gray dashed lines. **f** Average ion currents of the simulations depicted in panel (**e**). The "energy barrier" of potassium in each simulation is indicated in the text label. The Thr59 Oγ–Oγ distance is ~0.53 nm for all simulations. Results of the kinked state are not shown due to a different opening degree of the selectivity filter. Ion currents of the simulation replicates are shown as black dots. Each data point includes 10 independent replicates. **g** Representative conformations of the inner helix surrounding the open (kinked, white) and closed (bent, green) central cavity. Critical backbone hydrogen bonds are indicated by dotted lines. Dashed arrows indicate the rotation direction of the inner helix. Data are presented as mean values ± SEM. Source data are available as a Source Data file.

conformational rearrangement that involves both outer and inner helices, as well as the pore helix, as suggested by the FMA (Fig. 4a).

**Gating of the central cavity by Phe87 of the inner helix.** It has been reported that opening of the central cavity is involved in the activation of potassium channels[21]. For instance, bending of the inner helix opens the central cavity and allows ion permeation of KcsA[22]. In this regard, we checked whether the conformational change of the inner helix revealed by our simulations also modulates the opening of the central cavity of MthK. We observe that the bent state promotes dehydration of the central cavity, which becomes another barrier for ion conduction in the case of an efficiently dehydrated cavity at sufficiently high bent-state fraction.

Dehydration of the central cavity in the bent state is ascribed to the Phe87 side chains. We identified different orientations of the Phe87 side chain in the kinked and bent states in MthK. In the kinked state, it points toward adjacent subunits and leaves the

central cavity accessible to the bulk, whereas in the bent state, it resides in the central cavity so that the cavity is dehydrated and ion conduction is inhibited (Fig. 5a, b). We characterize the orientation of the Phe87 side chain by measuring the distance between the center of mass (COM) of the side chain and the COM of four Phe87 Cα atoms (referred to as Phe87-CoF distance). We expect a smaller Phe87-CoF distance if a Phe87 side chain points toward the central cavity. The histogram as a function of the Phe87-CoF and the Val81O–Gly85HN distance, which is used to discriminate the kinked and bent states of the inner helix, indicates the above-mentioned different Phe87 side-chain orientations in the bent and the kinked states (Fig. 5c). Correlation between Phe87 side-chain orientation and the inner helical conformation is also supported by the Phe87-CoF distance as a function of the bent-state fraction in simulations with and without restraints (Fig. 5d, Supplementary Fig. 6).

Dehydration of the central cavity by Phe87 side chains in the bent state constitutes an additional energy barrier for ion permeation (Fig. 5e, f) and serves as another gate of the channel. A series of simulations with different hydration degrees of the

central cavity, but similar opening degrees of the selectivity filter, prove this hypothesis: (a) the simulations with all four subunits restrained in the bent state show an average current of 2.9 pA, much smaller than expected, considering the large opening degree of the selectivity filter (Oγ–Oγ distance of 0.533 nm) (Supplementary Table 4). The smaller pore radius and higher energy barriers for the conduction of water and potassium in the central cavity in this bent state than in the kinked state (Fig. 5e) suggest dehydration of the central cavity, which may explain the reduced ion conduction rate. Applying additional restraints to the backbone hydrogen bonds of the helical turn immediately following the kink in the bent state (referred to as occluded state, Supplementary Tables 1 and 4, Fig. 5e, f) further stabilizes the Phe87 side-chain orientation in the cavity and promotes dehydration compared with the bent state (compare pore radius and "free energy" profiles in Fig. 5e) and results in a nonconductive channel (current of 0.7 pA, with a similar large Oγ–Oγ distance of 0.527 nm, Fig. 5f, Supplementary Table 4). (b) Mutation of Phe87 to Ala, a residue with a smaller side chain, results in a better hydrated central cavity and lower energy barrier for potassium than the wild type (2.6 and 5.1 $k_BT$, respectively; all four subunits are restrained in the bent state for both systems) (Fig. 5e, f). Simulations of this mutant show an opening degree of the selectivity filter (Oγ–Oγ distance of 0.529 nm) similar to the wild type, but with a much larger current (13.3 vs. 2.9 pA, respectively), in line with our hypothesis (Supplementary Table 4, Fig. 5f). (c) Restraining the Ile84 side-chain distance at 1.65 nm suggests an average current of 13.4 pA and an Oγ–Oγ distance of 0.538 nm (Supplementary Table 6, Supplementary Fig. 7). By looking at the distribution of the currents, the simulation replicas can be divided into two groups with similar average Thr59 Oγ–Oγ distances (0.537 vs. 0.541 nm) but different currents (6.9 vs. 25.4 pA, Supplementary Table 6). The group with a smaller average current reveals a higher bent-state fraction (71% vs. 54%) and a smaller Phe87-CoF distance (0.666 vs. 0.836 nm) than the group with a larger average current. The group with a higher bent-state fraction is accompanied with a smaller and less hydrated central cavity, and an increased energy barrier for potassium (Supplementary Fig. 7), consistent with our hypothesis. Analyses of the simulations with the Ile84 side-chain

distance restrained at 1.62 nm show similar results (Supplementary Fig. 7).

**Crosstalk between two gates of MthK.** In summary, a conformational change of the inner helix regulates ion permeation rates by modifying the structure of the permeation pathway. Specifically, the kink of the inner helix affects both the opening degree of the selectivity filter and the dehydration degree of the central cavity, which serve as two gates of MthK. Our model indicates a crosstalk between the two gates, in which opening of one gate finally leads to closure of the other. The channel is nonconductive due to a closed selectivity filter if all four subunits are in the kinked state. The channel is in another nonconductive state when all four subunits are in the bent state, and the central cavity is efficiently dehydrated (Fig. 6). Conductive states are found for asymmetrical structures when some subunits are in the kinked state, whereas the remaining ones are in the bent/occluded state. To further verify this hypothesis, we perform simulations with one, two, or three subunits restrained in the occluded state (the remaining ones are left unrestrained). Optimum ion current is found when two adjacent subunits are in the occluded state (~10.3 pA vs. ~5.4 and ~1.9 pA when one or three subunits in the occluded state, respectively, see Supplementary Table 4).

**Control simulations in the AMBER force field.** To test the robustness of our conclusions, as well as a possible dependence on the choice of force field in our case, we repeated some of the simulations using the AMBER14 force field for the protein and the Slipids force field for the lipids. These simulations confirm the above observations, suggesting reliability of our simulations (Supplementary Figs. 10–12, see Supplementary Note 6).

## Discussion
Our simulations identified a bent state of the inner helix of MthK (Figs. 3a, 4a, and 5g), as well as a conformational transition between the kinked and the bent state. The different ion permeation rates in different bilayers are the consequence of a shift of the conformational equilibrium between these two states due to lipid–protein interactions (Fig. 3c). This conformational

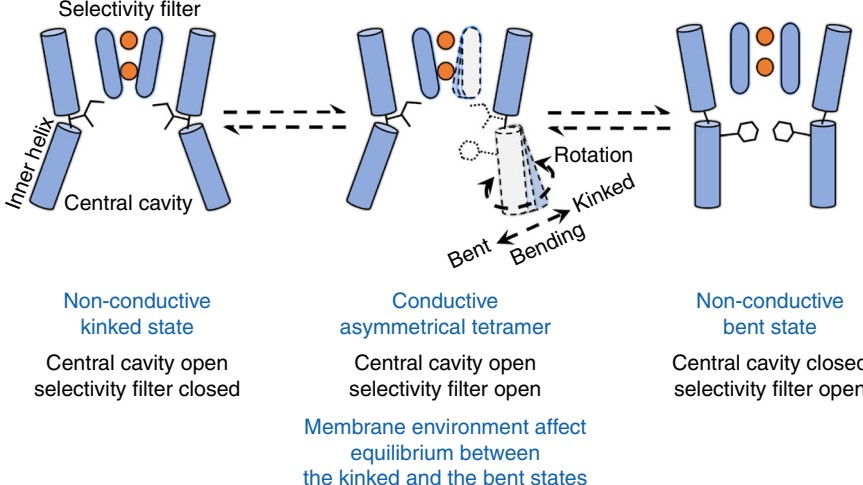

**Fig. 6 Hypothetical crosstalk between the two gates of MthK.** The channel is nonconductive due to closed selectivity filter if all four subunits are in the kinked state (left). Conformational transition from the kinked to the bent state opens the selectivity filter, but promotes dehydration of the central cavity (middle). The central cavity is efficiently dehydrated, and the channel is in another nonconductive state if all four subunits are in the bent state (right). The conductive state is found for asymmetrical tetrameric structures (middle). Lipid–protein interactions affect the conformational equilibrium between the kinked and bent states, which in turn regulates ion permeation rate. Directions of helix bending and rotation in the process of conformational transition are indicated by dashed arrows.

equilibrium suggested differential open probabilities of the pore domain of MthK in different membrane environments, which may be ascribed to different stabilities of the two states, or changed the energy barrier for the conformational transition between the kinked and the bent states. The conformation of the inner helix mediates a crosstalk between two gates at the selectivity filter and the central cavity, both of which contribute to the energy barrier of ion permeation (Figs. 4 and 5). Similar crosstalk between the central cavity and the selectivity has also been reported for KcsA[23]. This gating mechanism may also apply to other potassium channels.

Smaller ion permeation rates have also been reported by experiments for KcsA and the large-conductance $Ca^{2+}$-activated potassium channel ($BK_{Ca}$), a homolog of MthK, in thicker membranes[9–11], and in cholesterol-containing membranes[18,19] (see Supplementary Note 7). Our simulations are in agreement with these experimental results, and provide an atomistic view of how the lateral stress affects the open probability of the potassium channel in the case of MthK. We ascribe the lower open probability of MthK in thicker membranes and cholesterol-containing membranes in our simulations to the larger lateral pressure. The thicker membranes (e.g., DPPC, DGPC, DEPC, and DNPC) exert larger lateral pressure on the protein as the protein is buried more deeply in the hydrophobic region in thicker membranes, where the lateral pressure is larger. In contrast, in thinner membranes, the protein may reside in the vicinity of the headgroup–tail interface, where the lateral pressure is negative (Supplementary Fig. 13), and the forces exerted on the protein would be different (see Supplementary Note 7 for more discussions).

We rule out the effects of specific cholesterol–protein interactions in cholesterol-containing membranes, as we did not identify stable cholesterol-binding sites on protein. We also rule out protein conformational changes induced directly by the effects of thickness mismatch in the thicker membranes, because stabilization of the kinked state in these membranes decreased the hydrophobic region of the protein slightly and did not resolve the thickness mismatch. The thickness mismatch is mainly resolved by the distortion of the membranes in simulations, as Callahan et al.[11] found in their simulations of the KcsA potassium channel. However, the effects of thickness mismatch indeed apply in the case of DLPC, in which the hydrophobic region of the bilayer is smaller than that of the protein. The thinner membrane induced bending of the inner helix and promoted the shift of the conformational equilibrium to the kinked state.

Our analyses indicate that a higher unsaturation degree of lipid tails leads to a larger fraction of the kinked state. Previous studies suggested that lipids in close contact with the protein surface are less ordered than those distant from proteins[24], due to the irregular shape of the protein surface. A similar mechanism may apply to our case: less ordered tails of the more unsaturated lipids fit better with the kinked state, whose surface is more irregular than the bent state (as the inner helix is straighter), so that the conformational equilibrium is shifted to the kinked state. A rigorous test of this hypotheses requires more simulations, which is the scope of future work.

Our model suggests a gating mechanism of MthK, which involves a crosstalk between a gate at the selectivity filter and a gate at the central cavity: a kink of the inner helix, as well as the accompanying hinge-bending and rotation of this helix, modulates the two gates simultaneously but in opposite directions, so that opening of one gate eventually leads to closure of the other (Fig. 6). Thus, two nonconductive states are found due to a closed selectivity filter and a dehydrated central cavity, respectively (Fig. 6). The asymmetrical structure of the tetramer, in which some subunits are in the bent state, while the others are in the kinked state, enables a hydrated central cavity and an open selectivity filter simultaneously, and thus allows ion conduction (Figs. 3 and 6)[25]. We do not rule out the possibility that some intermediate states between the kinked and the bent state may allow ion permeation.

This model is consistent with Kopec et al.'s work[20], as both models suggest the opening degree of the selectivity filter as a gate of ion permeation. However, our model indicated that this gate at the selectivity filter was controlled by a kink, as well as the accompanying bending and rotation, of the inner helix, rather than the splay of these helices. In addition, dehydration of the central cavity in our simulations is similar with Jia et al.'s work[26], which indicated dewetting of the central cavity of the large-conductance, $Ca^{2+}$-gated (BK) potassium channel, a homolog of MthK. Our model further identified a coupling of the selectivity filter with the dehydration of the central cavity, both of which are results of the conformational changes of the inner helix. Multiple experimental results support our gating model of MthK. Constriction of the intracellular entrance of the central cavity has been proposed as a gating mechanism of some potassium channels such as KcsA[21,22,27]. However, in our model for MthK, only a slight narrowing of the intracellular entrance of the cavity was found, and it did not play the main role in gating (see pore radius in Fig. 5). The central cavity is mainly blocked by the side chain of Phe87 residues immediately below the selectivity filter (Fig. 5). This is in line with Posson et al.'s work[28,29], which suggested a narrowing, but not a constriction, of the central cavity in the closed state of MthK. Our model suggests two nonconductive states for MthK. Kuo et al.[30] indicated that MthK underwent desensitization after activation by $Ca^{2+}$. Experiments also suggested inactivation of MthK, which depends on the pH[30], voltage[31], and the $Ca^{2+}$[32] concentration of the environment. These observations imply the possible presence of multiple nonconductive states, and are in line with our model to some extent, although to assign each nonconductive state in our simulations to a specific state in experiments is beyond the scope of this work. We also note that this bidirectional coupling of a gate at the central cavity and a gate at the selectivity filter was also suggested for other potassium channels such as KcsA and Kv[12,23,33–36], although in these cases, a collapse or constriction of the selectivity filter is involved, rather than a simple closure of its intracellular entrance. The gating mechanism involves a hinge-bending of the inner helix in the vicinity of a glycine residue, which is conserved among different potassium channels (see Fig. 3 in ref. [21]). We therefore propose that our model may also apply to other potassium channels.

We studied ion permeation of MthK in different membrane environments with different lipid tails to assess the effects of lipid–protein interactions on potassium channel conformation and permeation using MD simulations. A variety of membrane properties (thickness, cholesterol content, and the unsaturation degree of lipid tails) were observed to affect the ion permeation rate. We assign this effect to a shift of the protein conformational equilibrium between a kinked and a bent state of the inner helix, which in turn modifies the structure of the permeation pathway, specifically the opening degree of the selectivity filter and the hydration degree of the central cavity, by modulating the positions and orientations of two key residue side chains: Ile84 and Phe87. The conformational changes revealed by our theoretical study also indicate a crosstalk between the hydration degree of the central cavity and the opening degree of the selectivity filter, implying that opening of the central cavity finally leads to closure of the selectivity filter and vice versa.

## Methods
**Molecular dynamics simulations.** Simulations in different membranes are summarized in Supplementary Table 2. MD simulations were performed using the

crystal structure of MthK (PDB entry: 3LDC)[17]. We employed phosphatidylcholine (PC) lipids with different tails (including DLPC 12:0/12:0, DMPC 14:0/14:0, DPPC 16:0/16:0, DYPC 16:1(n-9)/16:1(n-9), DOPC 18:1(n-9)/18:1(n-9), DGPC 20:1(n-11)/20:1(n-11), DEPC 22:1(n-13)/22:1(n-13), DNPC 24:1(n-15)/24:1(n-15), and POPC 16:0/18:1(n-9)) to investigate the effect of a variety of general membrane properties on ion permeation. These one-component membranes allowed us to test bilayer thicknesses (ranging from ~3.0 to ~5.0 nm) and unsaturation degree of lipid tails (saturated, unsaturated, and hybrid lipids). Binary mixtures of cholesterol and PC lipids (DLPC and POPC) with a cholesterol molar ratio of 33% were used to evaluate the effect of cholesterol content.

We also conducted a series of simulations with restraints on the protein to investigate how protein conformational changes revealed by our simulations are coupled to ion permeation. We also simulated the Ile84Ala and the Phe87Ala mutants, to assess the function of Ile84 and Phe87 in the ion permeation process. See Supplementary Methods for more details.

Molecular dynamics simulations were conducted using the GROMACS 5.1 software package[37] and the CHARMM36m force field[38]. Some of the simulations were repeated using the AMBER14 force field for the protein[39], the Slipids force field for the membranes[40–42], the TIP3P water model, and Joung[43] parameters for ions. CHARMM-GUI[44] was used to generate the simulation systems. Each system was subjected to a series of parallel production simulations with 10–40 simulation replicates and 0.5–1.5-μs simulation time for each replicate. The total simulation time was ~450 μs. A constant electrostatic field[45] to mimic a transmembrane voltage of 300 mV was applied (Fig. 1). The voltage $V$ was calculated by the following equation:

$$V = E \times L \tag{1}$$

where $E$ was the applied electrostatic field, and $L$ was the box size along the $z$ direction (the membrane normal direction). The electrostatic fields applied were ~0.0325 V·nm$^{-1}$. 1 mol·L$^{-1}$ potassium concentration that was used for all simulations. See the Supplementary Methods for details of the simulation protocol.

**Reporting summary**. Further information on research design is available in the Nature Research Reporting Summary linked to this article.

## Data availability

Data supporting the findings of this paper are available from the corresponding author upon reasonable request. A reporting summary for this article is available as a Supplementary Information file. The source data from MD simulations underlying Figs. 2, 3b–g, 4b, d–f, 5c, d, f, Supplementary Tables 2–7, and Supplementary Figs. 1–6, 7B, 8–12 are provided as a Source Data File available from https://doi.org/10.6084/m9.figshare.11827806.

## Code availability

The python scripts to calculate bending angle, distance, and the potassium and water energy profiles are available from https://doi.org/10.6084/m9.figshare.11827809.

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

## Acknowledgements
R.-X.G. is supported by the Alexander von Humboldt research fellowship for post-doctoral researchers. The authors thank Dr. Wojciech Kopec and Dr. Petra Kellers for insightful discussions and proofreading of the paper.

## Author contributions
B.L.d.G. designed and supervised the project. R.-X.G. performed molecular dynamics simulations and analyzed the data. R.-X.G. and B.L.d.G. wrote the paper.

## Competing interests
The authors declare no competing interests.
