## [Peer Review File · Nature Communications]

Reviewers' comments:

Reviewer #1 (Remarks to the Author):

In this study, the authors investigated the effect of different membrane environment on the channel permeation rates using molecular dynamics simulations. The authors found that the thickness of the bilayer, lipid unsaturation as well as cholesterol content all influence the ion permeation rate of MthK. Furthermore, from their restrained and unrestrained simulations, they identified two gates at the selectivity filter and the central cavity that are linked to each other. In the field of molecular dynamics, the influence of different membranes and lipids on the channel permeation has been questioned for quite some time. Therefore I highly appreciate this systematic investigation by the authors and I believe this study deserves publication in Nature Communications.

I have some minor suggestions and questions:

1. The authors showed a strong correlation between the Thr59 O(γ)-O(γ) distance and the Ile84-Thr59 side chain distance. Does the interaction between the Ile84-Thr59 also influence the ion occupancy at the S4 in the selectivity filter?
2. The authors stated that conductive states are found for asymmetrical structures when some subunits are in the kinked states whereas the remaining ones are in the bent state. Is the conductance the same when only one of the subunit is in the kinked state or two and three subunits are in the kinked state?
3. In a recent publication in Nature Communications (Hydrophobic gating in BK channels) a hydrophobic gate was identified in the pore. Since MthK has been considered as a bacterial homologue of the BK channel, do the authors think that both channels share in a way same mechanism in gating?
4. Is MthK a mechanosensitive channel? If the channel conductance and open probability are sensitive to the lateral pressure in the membrane as proposed by the authors, I would naively think that the channel would be also sensitive to the mechanical force. Is this the case?

Reviewer #2 (Remarks to the Author):

The paper demonstrates, by molecular dynamics computer simulations, that lipid membrane composition modulates conductivity of MthK potassium ion channel, and reveals molecular details of this phenomena. Particularly it was convincingly demonstrated that two states of the transmembrane helix near residues 81-86, "kink" and "bent", are related to lower respectively higher conductivity of the channel, while the equilibrium between these two states are affected by the composition of surrounding lipids. The study is very well designed and the text is clearly written. The main conclusion are supported by the use of different force fields showing similar results, as well as by repeating simulations significant amount of times. The study is also a nice example on how computer modeling can contribute to getting novel insight into membrane protein functioning. I favor publication of this paper, subject to some revision accountings for the points below.

1. The temperature of the simulations, as follows from the SI, was set to 323 K. I guess the reason for this was to provide conditions for some lipids (particularly DPPC) to be in a liquid crystalline phase, but question arise how the temperature difference (compared to the physiological temperature) may affect results of the paper. The authors need to discuss possible effects of the temperature shift.

2. More details should be given of how the initial state of the system was prepared and equilibrated. E.g., whether protein was fixed while lipids and water were equilibrated, when the pressure coupling was switched on, etc. Also, whether the same initial state was used in all replicas of the same system?

3. What was the electric field applied to the system (in units like V/m)? It is not enough to say "to mimic a transmembrane voltage of 300 mV" since this can be interpreted in different ways.

4. Location of double bond in unsaturated lipids should be given (like POPC 16:0/18:1(n-9))

Reviewer #3 (Remarks to the Author):

In this manuscript, the authors describe the regulation of MthK channel ion conductance by a variety of membrane lipids. The experiments were well performed, and I particularly appreciated the validation of results by using 2 unrelated force-fields. The authors conclude that in MthK, membrane thickness alters ion conductance in part through affecting the equilibrium of bent to kinked conformations of the inner helix. The authors' data suggests that kinking of this inner helix affects both the opening degree of the selectivity filter and the dehydration degree of the central. The authors propose specific atomic interactions that enable the inner helix to impact the structure of the selectivity filter. Therefore, the authors provide specific molecular details about how the lipid conditions examined can alter ion conductance in MthK channels.

Overall, this paper provides novel insights into the molecular details of lipid regulation of MthK channels. Experiments are appropriate, detailed, and well executed. The manuscript is well written.

I offer only these minor points to address.

Minor Points:

Lines 111 – 116: Results on Cholesterol. Since cholesterol can have multiple effects on protein function through both changes in bilayer properties and through direct interactions with the protein, it would be useful to briefly mention in this paragraph whether or not cholesterol was interacting with any of the channel subunits in any of your simulations, and whether or not such interactions may contribute to the observed changes in MthK's conductance in that group of simulations. The authors do finally address in the section of lines 387 – 388, however, there it is somewhat of a late display of important results.

Line 12: "Membranes do not only provide a matrix for a variety..." should say "Membranes not only provide a matrix for a variety..."

Lines 52 and 53: Did you mean "responds" instead of "response" in this sentence "However, it remains unclear how the channel structure response to the membrane environment at the atomistic level."??

Line 53: The sentence "This is where MD simulations can provide the missing information." seems a bit awkward. Perhaps something like "This is where MD simulations can provide valuable insights."

Lines 75-77: This sentence is poorly structured. Please reword. "More importantly, the conformational changes of the inner helix revealed by our simulations define two gates at the selectivity filter and the central cavity, respectively, and mediate a crosstalk between these two gates, suggesting a complex way of channel gating."

Line 77: "Our work provided" should say "Our work provides"

Line 392: "The thickness mismatch is mainly resolved by the distortion of the membranes in simulations." I think it's worth noting that this is similar to what was observed for hydrophobic mismatch in KcsA by Callahan et al., 2019.

Supp. Doc.

Line 33. "We conducted a series of simulations with restraints..." should say "We conducted a series of simulations of the MthK potassium channel with restraints..."

Reviewers' comments:

Reviewer #1 (Remarks to the Author):

In this study, the authors investigated the effect of different membrane environment on the channel permeation rates using molecular dynamics simulations. The authors found that the thickness of the bilayer, lipid unsaturation as well as cholesterol content all influence the ion permeation rate of MthK. Furthermore, from their restrained and unrestrained simulations, they identified two gates at the selectivity filter and the central cavity that are linked to each other. In the field of molecular dynamics, the influence of different membranes and lipids on the channel permeation has been questioned for quite some time. Therefore, I highly appreciate this systematic investigation by the authors and I believe this study deserves publication in Nature Communications.

I have some minor suggestions and questions:

1. The authors showed a strong correlation between the Thr59 O(γ)-O(γ) distance and the Ile84-Thr59 side chain distance. Does the interaction between the Ile84-Thr59 also influence the ion occupancy at the S4 in the selectivity filter?

Reply: We thank the reviewer for the constructive suggestions. We indeed found a correlation between Ile84-Thr59 side chain distance and the ion occupancy at S4 and S3 sites in our simulations. However, we propose that this is mainly because the Ile84-Thr59 side chain interactions modulate the Thr59 O γ -O γ distance, which in turn regulates ion occupancy. Supplementary Fig. 5 indicated that a larger O γ -O γ distance associates with a lower ion occupancy at S4 site and a higher occupancy at S3 site. We discussed this point at the end of the “Opening of the selectivity filter in the two states of MthK” section.

2. The authors stated that conductive states are found for asymmetrical structures when some subunits are in the kinked states whereas the remaining ones are in the bent state. Is the conductance the same when only one of the subunit is in the kinked state or two and three subunits are in the kinked state?

Reply: The ion permeation rates are not the same when different numbers of subunits are in the kinked state.

First, this can be seen from Fig. 3C, which showed different currents for MthK with different bent state fraction. In addition, we performed simulations in which one, two, three, and all four subunits are restrained in the occluded state, respectively, while the remaining subunits are left free. We restrained the subunits in the occluded state (see Supplementary Table 1) so that (a) Ile84 side chain was moved away from Thr59 and (b) Phe87 side chain pointed toward the central cavity. The subunits which are not restrained remained in the kinked state during most of the simulation time. When more subunits restrained in the occluded state, larger opening degree

of the selectivity filter is found, but the central cavity is more dehydrated. The optimum current was observed when two subunits were restrained in the occluded state (~10 pA), while ion permeation rates are much smaller when one or three subunits are restrained in the occluded state (~5 pA, and ~2 pA). The channel is non-conductive if all four subunits are in the occluded state. These results are summarized in Supplementary Table 4 and at the end of the “*Gating of the central cavity by Phe87 of the inner helix*” section of the manuscript.

3. In a recent publication in Nature Communications (Hydrophobic gating in BK channels) a hydrophobic gate was identified in the pore. Since MthK has been considered as a bacterial homologue of the BK channel, do the authors think that both channels share in a way same mechanism in gating?

Reply: We agree with the reviewer that BK channel may share the same mechanism. Actually, BK channels are what we are planning to simulate as a follow-up project. In the revised manuscript, we discussed the results of BK channel simulations by Jia et al. in the last paragraph of the discussion section.

4. Is MthK a mechanosensitive channel? If the channel conductance and open probability are sensitive to the lateral pressure in the membrane as proposed by the authors, I would naively think that the channel would be also sensitive to the mechanical force. Is this the case?

Reply: A mechanosensitive channel refers to a channel gated by the mechanical force of the membrane. Our simulations indicated that the equilibrium between two states of the channel is affected by the lateral pressure of the bilayer. Our results may suggest that the lateral pressure is able to modulate the open probability of the channel in the presence of a gating signal. However, our simulations are not able to tell whether the lateral pressure is a gating signal of the channel. In this regard, we cannot conclude whether MthK is a mechanosensitive channel or not and hence prefer not to speculate in that context.

Reviewer #2 (Remarks to the Author):

The paper demonstrates, by molecular dynamics computer simulations, that lipid membrane composition modulates conductivity of MthK potassium ion channel, and reveals molecular details of this phenomena. Particularly it was convincingly demonstrated that two states of the transmembrane helix near residues 81-86, "kink" and "bent", are related to lower respectively higher conductivity of the channel, while the equilibrium between these two states are affected by the composition of surrounding lipids. The study is very well designed and the text is clearly written. The main conclusion are supported by the use of different force fields showing similar results, as well as by repeating simulations significant amount of times. The study is also a nice example on how computer modeling can contribute to getting novel insight into membrane protein functioning. I favor publication of this paper, subject to some revision accountings for the points below.

1. The temperature of the simulations, as follows from the SI, was set to 323 K. I guess the reason for this was to provide conditions for some lipids (particularly DPPC) to be in a liquid crystalline phase, but questions arise how the temperature difference (compared to the physiological temperature) may affect results of the paper. The authors need to discuss possible effects of the temperature shift.

Reply: Keeping the membrane in the liquid phase state is indeed one reason of using a higher temperature. Another effect is that the higher temperature accelerates conformational transitions in MD simulations, which is essential for sampling as much conformational transition events as possible at given simulation time. This also applies to the observed ion current, our primary and limiting readout from the simulations. We do not expect any adverse effects on the protein as MthK is extracted from *Methanothermobacter thermautotrophicus*, a thermophile of which the maximum growth rate lies even at a slightly higher temperature. We discussed these points in the section of “*Parameters of molecular dynamics simulations*” in the supplementary information.

2. More details should be given of how the initial state of the system was prepared and equilibrated. E.g., whether protein was fixed while lipids and water were equilibrated, when the pressure coupling was switched on, etc. Also, whether the same initial state was used in all replicas of the same system?

Reply: We first equilibrated the system with gradually removed restraints on the protein and lipids in six steps using the default scheme suggested by CHARMM-GUI. We then performed 0.1-0.3 μ s simulations without any restraints. The electric field was not applied at this stage. At last we conducted production simulations with transmembrane voltage applied. For the production simulation, we randomly selected 2-4 snapshots from the equilibrium simulations as initial conformation, depending on the number of replicates we simulated. We included these details in the section of “*Parameters of molecular dynamics simulations*” in the supplementary information.

3. What was the electric field applied to the system (in units like V/m)? It is not enough to say "to mimic a transmembrane voltage of 300 mV" since this can be interpreted in different ways.

Reply: We applied an electric field of ~ 0.0325 V/nm. The transmembrane voltage V was calculated by the following equation:

$$V = E \times L$$

where E is the electric field strength and L is the box size along the z direction. Because of the low dielectric of the membrane, the vast majority of the applied voltage drops across the membrane. We used approximately the same box size for different systems, so that only minor change of the electric field strength ($<3\%$) is necessary to result in the same transmembrane voltage. We explained this point in the Method section in the revision.

4. Location of double bond in unsaturated lipids should be given (like POPC 16:0/18:1(n-9))

Reply: We noted the locations of double bonds in unsaturated lipids in the Method section of the manuscript and in Table S2 and S7 of the supplementary information.

Reviewer #3 (Remarks to the Author):

In this manuscript, the author's describe the regulation of MthK channel ion conductance by a variety of membrane lipids. The experiments were well performed, and I particularly appreciated the validation of results by using 2 unrelated force-fields. The authors conclude that in MthK, membrane thickness alters ion conductance in part through affecting the equilibrium of bent to kinked conformations of the inner helix. The authors' data suggests that kinking of this inner helix affects both the opening degree of the selectivity filter and the dehydration degree of the central. The authors propose specific atomic interactions that enable the inner helix to impact the structure of the selectivity filter. Therefore, the authors provide specific molecular details about how the lipid conditions examined can alter ion conductance in MthK channels.

Overall, this paper provides novel insights into the molecular details of lipid regulation of MthK channels. Experiments are appropriate, detailed, and well executed. The manuscript is well written.

I offer only these minor points to address.

Minor Points:

Lines 111 – 116: Results on Cholesterol. Since cholesterol can have multiple effects on protein function through both changes in bilayer properties and through direct interactions with the protein, it would be useful to briefly mention in this paragraph whether or not cholesterol was interacting with any of the channel subunits in any of your simulations, and whether or not such interactions may contribute to the observed changes in MthK's conductance in that group of simulations. The authors do finally addressed in the section of lines 387 – 388, however, there it is somewhat of a late display of important results.

Reply: In the revised version, we mentioned in this paragraph that cholesterol are depleted from MthK during the simulations, and proposed the possibility that cholesterol modulate ion conduction by changing the lateral pressure of the protein.

Line 12: "Membranes do not only provide a matrix for a variety..." should say "Membranes not only provide a matrix for a variety..."

Lines 52 and 53: Did you mean "responds" instead of "response" in this sentence "However, it

remains unclear how the channel structure response to the membrane environment at the atomistic level.”??

Line 53: The sentence “This is where MD simulations can provide the missing information.” seems a bit awkward. Perhaps something like “This is where MD simulations can provide valuable insights.”

Reply: The above three corrections suggested by the reviewer are included in the revised manuscript.

Lines 75-77: This sentence is poorly structured. Please reword. “More importantly, the conformational changes of the inner helix revealed by our simulations define two gates at the selectivity filter and the central cavity, respectively, and mediate a crosstalk between these two gates, suggesting a complex way of channel gating.”

Reply: We rephrased the sentence in the revision as “More importantly, the conformational changes of the inner helix revealed by our simulations define two gates at the selectivity filter and the central cavity, respectively. We also revealed a crosstalk between these two gates mediated by the conformational changes, suggesting a complex way of channel gating.”

Line 77: “Our work provided” should say “Our work provides”

Reply: We revised the manuscript accordingly.

Line 392: “The thickness mismatch is mainly resolved by the distortion of the membranes in simulations.” I think it’s worth noting that this is similar to what was observed for hydrophobic mismatch in KcsA by Callahan et al., 2019.

Reply: We noted the match between our simulations and Callahan et al.’s work accordingly in the revision.

Supp. Doc.

Line 33. “We conducted a series of simulations with restraints...” should say “We conducted a series of simulations of the MthK potassium channel with restraints...”

Reply: We corrected the supporting information accordingly.

REVIEWERS' COMMENTS:

Reviewer #1 (Remarks to the Author):

All questions are answered satisfactory. I have no further comment and therefore recommend this paper to be published in Nature Communications.

Reviewer #2 (Remarks to the Author):

In the revised manuscript the authors addressed to all reviewer comments, and I can recommend the paper for publication

Reviewer #3 (Remarks to the Author):

I am satisfied with all changes made to the manuscript. Congratulations on an excellent study.

EXTENDED COMMENTS: NCOMMS-19-40919A .

Data presentation: Please ensure that data presented in a plot, chart or other visual representation format shows data distribution clearly (e.g. dot plots, box-and-whisker plots). When using bar charts, please overlay the corresponding data points (as dot plots) whenever possible and always for $n \leq 10$. (Please see the following editorial for the rationale behind this request and an example <https://www.nature.com/articles/s41551-017-0079>).

Please note that data presentation has to be revised to comply with our policy in figures 5f; S7b.

Reply: We overlaid the corresponding data points as dots in Fig. 5F, and Supplementary Fig. S7B in the revised version.

Statistics: Wherever statistics have been derived (e.g. error bars, box plots, statistical significance) the legend needs to provide and define the n number (i.e. the sample size used to derive statistics) as a precise value (not a range), using the wording “n=X biologically independent samples/animals/cells/independent experiments/n= X cells examined over Y independent experiments” etc. as applicable.

Please note that this information is missing in the figure legends of 2; 3c-3g; 4b, 4d, 4f-4g; 5f; S1; S3a-S3c; S4; S5; S6; S7b; S10c; S11a-S11b, S11d-S11e; S12 (Right Panel).

Statistics such as error bars cannot be derived from $n < 3$ and must be removed from all such cases.

We strongly discourage deriving statistics from technical replicates, unless there is a clear scientific justification for why providing this information is important. Conflating technical and biological variability, e.g., by pooling technically replicates samples across independent experiments is strongly discouraged. (For examples of expected description of statistics in figure legends, please see the following <https://www.nature.com/articles/s41467-019-11636-5> or <https://www.nature.com/articles/s41467-019-11510-4>).

All error bars need to be defined in the legends (e.g. SD, SEM) together with a measure of centre (e.g. mean, median). For example, the legends should state something along the lines of “Data are presented as mean values +/- SEM” as appropriate.

All box plots need to be defined in the legends in terms of minima, maxima, centre, bounds of box and whiskers and percentile.

The figure legends must indicate the statistical test used. Where appropriate, please indicate in the figure legends whether the statistical tests were one-sided or two-sided and whether adjustments were made for multiple comparisons.

For null hypothesis testing, please indicate the test statistic (e.g. F, t, r) with confidence intervals, effect sizes, degrees of freedom and P values noted.

Please provide the test results (e.g. P values) as exact values whenever possible and with confidence intervals noted.

Reply: We modified the figure legends accordingly in the revision.

TITLE PAGE

* Please ensure that all affiliations are in the correct sequential order according to their position in the author list. Affiliation 1 must be associated with the first author. Please see this article for further detail: <https://www.nature.com/articles/s41467-018-04254-0.pdf>

we need affiliation numbers, even if authors share the same number/affiliation.

Reply: We added affiliation number on the title page of the revision.

MAIN TEXT

* Please shorten the main manuscript text (Introduction, Results, and Discussion, not including figure legends or Methods) to approximately 5,000 words or fewer.

Reply: We modified the manuscript accordingly, and the revised version contains ~5050 words. Specifically, we moved part of the results and discussions sections to the supplementary information.

* We allow only one level of subheadings in the Results section. Please remove secondary subheadings (see results section and methods section).

Reply: We removed the second level of subheadings in the Results section in the revision. We also revised the left subheadings so that the readability of the manuscript will not be affected.

* Please remove the subheadings from the Discussion section.

Reply: We do not have subheadings in the Discussion section in the current version.

LANGUAGE AND STYLE

* Please remove language such as "new", "novel", "for the first time", "unprecedented", etc. Novelty should be clear from the context.

Reply: We removed these words in the revised version.

* We do not allow inferences based on data that is not present in the manuscript or not published. Please include all the data that is not shown or change the statements that pertain to this data.

Reply: We do not have data that is not shown in the manuscript.

* Please do not use italics or bold font to convey emphasis (in both the main text and the display items).

Reply: We did not use italics and bold font in the revised version.

* Please avoid using speech marks around words or phrases. In most cases they are unnecessary.

Reply: We avoided using speech marks as possible as we can in the revision. The only exception is when we mention free energy profiles.

* Please make sure that mathematical terms throughout your manuscript and Supplementary Information (including in figures, figure axes, and legends) conform strictly to the following guidelines. Equations should be supplied in editable format, and not as images. Scalar variables (e.g. x , V , χ) should be typeset in italic, whereas multi-letter variables should be formatted in roman. Constants (e.g. \hbar , G , c) should be typeset in italics (the only exceptions being e , i , π , which should be typeset in Roman) and vectors (such as r , the wavevector k , or the magnetic field vector B) should be typeset in bold without italics. In contrast, subscripts and superscripts should only be italicised if they too are variables or constants. Those that are labels (such as the 'c' in the critical temperature, T_c , the 'F' in the Fermi energy, E_F , or the 'crit' in the critical current, I_{crit}) should be typeset in roman. To avoid doubt, unit dimensions should be expressed using negative integers (e.g. $\text{kg m}^{-1} \text{s}^{-2}$, not kg/ms^2) or the word 'per'.

Reply: We revised accordingly in the main text and the supplementary information.

* Please label equations sequentially as (1), (2), (3), etc. Subdivisions (1a, 1b, or 1.1, 1.2) are not permitted.

Reply: We labeled the equation accordingly in the method section in the revision.

METHODS AND DATA

* All *Nature Communications* manuscripts must include a section titled "Data Availability" as a separate section after the Methods section and before the References. For more information on this policy, and a list of examples, please see <http://www.nature.com/authors/policies/data/data-availability-statements-data-citations.pdf>

Reply: We included the data availability section in the manuscript.

* DATA SOURCES: Nature Research policies strongly encourage deposition of research data in public repositories and in some cases this is mandatory, and you may have been previously advised if that was the case. If you need help depositing and curating your research data

(including raw and processed data, text, video, audio and images) you should consider:
Contacting Springer Nature's Research Data Helpdesk for advice
Finding a suitable data repository for your data
Uploading your data to Springer Nature's Research Data Support service
Research Data Support is an optional Springer Nature service. There are fees for using this service, however, if you receive funding from the Wellcome Trust or are affiliated to a Wellcome Centre you can use Research Data Support at no cost. See here for more information. Please provide a unique identifier for the data (for example a DOI or a permanent URL) in the data availability statement, if possible. If the repository does not provide identifiers, we encourage authors to supply the search terms that will return the data. For data that have been obtained from publicly available sources, please provide a URL and the specific data product name in the data availability statement. Data with a DOI should be included in the reference list and cited where relevant.
Alternatively, include the data in the Supplementary Information. For datasets for which mandatory deposition is not required and the data can only be shared on request, please explain why in your Data Availability Statement and in your cover letter.
Please refer to our data policies here: <http://www.nature.com/authors/policies/availability.html>

Reply: We deposited our data in Figshare and provided a permanent URL in the data availability statement.

* We strongly encourage authors to deposit all code associated with the paper in a persistent repository where they can be freely and enduringly accessed. For all studies developing new software or using custom code that is deemed central to the conclusions, a statement must be included, under the heading "Code Availability", indicating whether and how the code can be accessed, including any restrictions to access. If the code can only be shared on request, please explain why in your Code Availability Statement and in your cover letter.

Reply: We deposited our scripts in Figshare and a permanent URL is included in the code availability statement.

DISPLAY ITEMS

* Please check whether your manuscript or Supplementary Information contain third-party images, such as figures from the literature, stock photos, clip art or commercial satellite and map data. We strongly discourage the use or adaptation of previously published images, but if this is unavoidable, please request the necessary rights documentation to re-use such material from the relevant copyright holders and return this to us when you submit your revised manuscript.

Reply: We did not use previously published images in our manuscript.

* Please ensure that figure legend titles are brief - they should not occupy more than one line in

the final proof.

Reply: We double checked the figure legend titles in the manuscript and we think they are brief.

* Please ensure that all colour scales are defined in either the figure or its associated legend.

Reply: All colour scales are defined in the corresponding figures.

* Please define any new abbreviations, symbols or colours present in your figures in the associated legends. Please do not use symbols in your legend, instead please write out the symbols in words (blue circles, red dashed line, etc.).

Reply: We double checked our manuscript and we believe the figure legends comply with the above guidelines.

* In each figure and supplementary figure where error bars are used, they must be defined. One statement at the end of each figure is sufficient if the error bars are equivalent throughout the figure.

Reply: We indicated at the end of the figure legends that the error bars are standard error of the mean (s.e.m.).

SUPPLEMENTARY INFORMATION

* We do not edit Supplementary Information files; they will be uploaded with the published article as they are submitted with the final version of your manuscript. Any tracked changes should be removed from the file and the file should be provided as a PDF file. Supplementary Figures do not need to be provided separately.

Reply: All of the tracked changes are removed form the supplementary information files and a PDF file is submitted.

* Supplementary References should appear at the end of the Supplementary Information file, and should be self-contained and numbered from 1. References mentioned in both the main text and the Supplementary Information should be part of both reference lists so that the Supplementary Information does not refer to the reference list in the main paper and vice versa.

Reply: We moved the Supplementary References to the end of the Supplementary Information file. We double checked the reference section and we believe it complies with the above guidelines.

* An updated editorial policy checklist that verifies compliance with all required editorial

policies must be completed and uploaded as a related manuscript file with the revised manuscript. All points on the policy checklist must be addressed; if needed, please revise your manuscript in response to these points. Please note that this form is a dynamic "smart pdf" and must therefore be downloaded and completed in Adobe Reader, instead of opening it in a web browser. Editorial policy checklist: <https://www.nature.com/authors/policies/Policy.pdf>

Reply: We submitted an updated editorial policy checklist.

* An updated reporting summary must be completed and uploaded as a supplementary information file with the revised manuscript. All points on the reporting summary must be addressed; if needed, please revise your manuscript in response to these points. Please note that this form is a dynamic "smart pdf" and must therefore be downloaded and completed in Adobe Reader, instead of opening it in a web browser.

Reporting summary: <https://www.nature.com/authors/policies/ReportingSummary.pdf>

* Reporting guidelines: Please find attached a reporting summary that includes comments on how to revise it in line with our policies and requests the addition of further information in the text when needed. An updated reporting summary must be completed and uploaded as a supplementary information file with the revised manuscript. This checklist is published alongside your manuscript online.

Please also find attached a Word document containing requests for additional information in the figure legends, text, and Methods section to comply with our reporting policies.

Reply: We revised the reporting summary according to the comments and guidelines.

We note that we did not change the "Data" section of the reporting summary. In the comment, it was suggested to include the PDB entry of the protein structure. However, we just used it as the initial conformation in our MD simulations. The structure is not a result of this work. In this regard, we chose not to include the PDB entry.

The issues in the extended comment in the word document are addressed, as replied on the first page of this document.

* Your paper will be accompanied by a two-sentence Editor's summary, of between 250-300 characters including spaces, when it is published on our homepage. Could you please approve the draft summary below or provide us with a suitably edited version.

EDITOR'S SUMMARY

Potassium (K⁺) channels, such as MthK, are essential for many biological processes, but how lipid-protein interactions regulate ion permeation of K⁺ channels remained unclear. Here authors conducted molecular dynamics simulations of MthK and observed different ion permeation rates

of MthK in membranes with different properties.

Reply: Thanks for drafting this summary. It works for us.

* As part of our efforts to communicate our content to a wider audience, we endeavour to highlight papers published in *Nature Communications* on the journal's Twitter account ([@NatureComms](https://twitter.com/NatureComms)). If you would like us to mention authors, institutions or lab groups in these tweets, please provide the relevant twitter handles in your cover letter upon resubmission.

Reply: We provided the twitter handle of the institution (@CompBioPhys) in the cover letter.

* If you opted into the journal hosting details of a preprint version of your manuscript via a link on our dedicated website (<https://nature-research-under-consideration.nature.com>), it will remain on this site while you are revising your manuscript, as we consider the file to remain active. Should you wish to remove these details, please email naturecommunications@nature.com indicating your manuscript number and the link on our website that was previously sent to you. Please see our pre-publicity policy at <http://www.nature.com/authors/policies/confidentiality.html> For more information, please refer to our FAQ page at <https://nature-research-under-consideration.nature.com/posts/19641-frequently-asked-questions>

Reply: We did not release a preprint of the manuscript.

* In recognition of the time and expertise our reviewers provide to *Nature Communications*'s editorial process, as of November, 2018, we formally acknowledge their contribution to the external peer review of articles published in the journal. All peer-reviewed content will carry an anonymous statement of peer reviewer acknowledgement, and for those reviewers who give their consent, we will publish their names alongside the published article. For more information, please refer to our FAQ page at <https://www.nature.com/documents/ncomms-reviewer-information.pdf>

Reply: Thanks for this message. We aware of this policy.

OPEN ACCESS:

Nature Communications is a fully open access journal. Articles are made freely accessible on publication under a [CC BY license](https://creativecommons.org/licenses/by/4.0/) (Creative Commons Attribution 4.0 International License). This license allows maximum dissemination and re-use of open access materials and is preferred by many research funding bodies.

For further information about article processing charges, open access funding, and advice and support from Nature Research, please visit <http://www.nature.com/ncomms/about/open-access>

SUBMISSION INFORMATION:

In order to accept your paper, we require the following:

- * A cover letter describing your response to our editorial requests.

Reply: We provided a cover letter according to your request.

- * The final version of your text as a Word or TeX/LaTeX file, with any tables prepared using the Table menu in Word or the table environment in TeX/LaTeX and using the 'track changes' feature in Word.

Reply: We submitted a word file with track changes.

- * The complete author list provided in the article file, which must match that given on our manuscript tracking system. The author list in the main article file will be used during typesetting of your article.

Reply: We confirmed the match between the author list in the article file and that on the manuscript tracking system.

- * Production-quality versions of all figures, supplied as separate files containing all panels. To ensure the swift processing of your paper please provide the highest quality, vector format, versions of your images (.ai, .eps, .psd) where available. Please see our brief guide to manuscript submission for further details on the figure formats we can accept. Text and labelling should be in a separate layer to enable editing during the production process. If vector files are not available then please supply the figures in whichever format they were compiled in and not saved as flat .jpeg or .TIFF files. Any chemical structures or schemes contained within figures should additionally be supplied as separate ChemDraw (.cdx) files. If your artwork contains any photographic images, please ensure these are at least 300 dpi.

To ensure that your figures are accessible to colour-blind readers, we encourage you to use alternative colour schemes. For example, rainbow colour scales may be replaced by single-colour intensity scales or greyscale, and red/green image overlays may be replaced with magenta/green. For reference an example of R-script colour blindness palettes can be found here <https://cran.r-project.org/web/packages/viridis/vignettes/intro-to-viridis.html>. Another example for Python can be found here: <http://matplotlib.org/cmocan/>

Reply: We checked our figures according to the above policies and submitted required images as separated files

- * The final version of any Supplementary Information (figures, tables, notes etc) in one PDF file. Please add a cover page to the Supplementary Information PDF, including the title of the

manuscript and the first author's surname in the format 'Smith et al.' Please submit movies, audio files and data sets as separate files. See <http://www.nature.com/ncomms/submit/how-to-submit#Supplementary-information> for acceptable file formats/sizes.

** Please note that Supplementary Information must be finalised prior to acceptance of the paper.
**

Reply: We revised the supplementary information accordingly and submitted a single PDF file for the supplementary information.

* If you wish, an interesting image (but not an illustration or schematic) for consideration as a 'Featured Image' on the Nature Communications homepage. Examples can be seen on our Facebook page: <http://go.nature.com/PGPizM> The file should be 1400x400 pixels in RGB format and should be uploaded as 'Related Manuscript File'. In addition to our home page, we may also use this image (with credit) in other journal-specific promotional material.

Reply: We decided not to provide the Featured Image.

* A completed author checklist, uploaded as a Related Manuscript file type, available at: <https://www.nature.com/documents/ncomms-manuscript-checklist.pdf>

Reply: We checked our manuscript to make it comply with requirements in the checklist.

* Completed and signed copies of our Multimedia License to Publish (LTP) for any Featured Image suggestions (please use one form for each image and give a scientific description of the image in the 'title' field; do not use "Featured Image" as a title):
Multimedia Licence to Publish form

Reply: We do not provide a Featured Image, so this does not apply to us.

At acceptance, the corresponding author will be required to complete an Open Access Licence to Publish on behalf of all authors, declare that all required third party permissions have been obtained and provide billing information in order to pay the article-processing charge (APC) via credit card or invoice.

Please note that your paper cannot be sent for typesetting to our production team until we have received these pieces of information; **therefore, please ensure that you have this information ready when submitting the final version of your manuscript.**

Reply: We aware of this policy.

ORCID

Nature Communications is committed to improving transparency in authorship. As part of

our efforts in this direction, we are now requesting that all authors identified as ‘corresponding author’ create and link their Open Researcher and Contributor Identifier (ORCID) with their account on the Manuscript Tracking System (MTS) prior to acceptance. ORCID helps the scientific community achieve unambiguous attribution of all scholarly contributions. For more information please visit

<http://www.springernature.com/orcid>

For all corresponding authors listed on the manuscript, please follow the instructions in the link below to link your ORCID to your account on our MTS before submitting the final version of the manuscript. If you do not yet have an ORCID you will be able to create one in minutes.

IMPORTANT: All authors identified as ‘corresponding author’ on the manuscript must follow these instructions. Non-corresponding authors do not have to link their ORCIDs but are encouraged to do so. Please note that it will not be possible to add/modify ORCIDs at proof. Thus, if they wish to have their ORCID added to the paper they must also follow the above procedure prior to acceptance.

To support ORCID's aims, we only allow a single ORCID identifier to be attached to one account. If you have any issues attaching an ORCID identifier to your MTS account, please contact the Platform Support Helpdesk.

Reply: We linked our ORCIDs to our accounts.